# Adapting Neural Link Predictors
# for Data-Efficient Complex Query Answering

**Erik Arakelyan**[*1]     **Pasquale Minervini**[*†2]     **Daniel Daza**[3,4,5]
**Michael Cochez**[3,5]     **Isabelle Augenstein**[1]
[1]University of Copenhagen     [2]University of Edinburgh     [3]Vrije Universiteit Amsterdam
[4]University of Amsterdam     [5]Discovery Lab, Elsevier, The Netherlands
{erik.a,augenstein}@di.ku.dk p.minervini@ed.ac.uk {d.dazacruz,m.cochez}@vu.nl

## Abstract

Answering complex queries on incomplete knowledge graphs is a challenging task where a model needs to answer complex logical queries in the presence of missing knowledge. Prior work in the literature has proposed to address this problem by designing architectures trained end-to-end for the complex query answering task with a reasoning process that is hard to interpret while requiring data and resource-intensive training. Other lines of research have proposed re-using simple neural link predictors to answer complex queries, reducing the amount of training data by orders of magnitude while providing interpretable answers. The neural link predictor used in such approaches is not explicitly optimised for the complex query answering task, implying that its scores are not calibrated to interact together. We propose to address these problems via CQD$^{\mathcal{A}}$, a parameter-efficient score *adaptation* model optimised to re-calibrate neural link prediction scores for the complex query answering task. While the neural link predictor is frozen, the adaptation component – which only increases the number of model parameters by $0.03\%$ – is trained on the downstream complex query answering task. Furthermore, the calibration component enables us to support reasoning over queries that include atomic negations, which was previously impossible with link predictors. In our experiments, CQD$^{\mathcal{A}}$ produces significantly more accurate results than current state-of-the-art methods, improving from $34.4$ to $35.1$ Mean Reciprocal Rank values averaged across all datasets and query types while using $\leq 30\%$ of the available training query types. We further show that CQD$^{\mathcal{A}}$ is data-efficient, achieving competitive results with only $1\%$ of the complex training queries, and robust in out-of-domain evaluations. Source code and datasets are available at https://github.com/EdinburghNLP/adaptive-cqd.

## 1   Introduction

A Knowledge Graph (KG) is a knowledge base representing the relationships between entities in a relational graph structure. The flexibility of this knowledge representation formalism allows KGs to be widely used in various domains. Examples of KGs include general-purpose knowledge bases such as Wikidata [Vrandečić and Krötzsch, 2014], DBpedia [Auer et al., 2007], Freebase [Bollacker et al., 2008], and YAGO [Suchanek et al., 2007]; application-driven graphs such as the Google Knowledge Graph, Microsoft's Bing Knowledge Graph, and Facebook's Social Graph [Noy et al., 2019]; and domain-specific ones such as SNOMED CT [Bodenreider et al., 2018], MeSH [Lipscomb, 2000], and Hetionet [Himmelstein et al., 2017] for life sciences; and WordNet [Miller, 1992] for linguistics.

---

*Equal contribution, alphabetical order.   †Senior author.

37th Conference on Neural Information Processing Systems (NeurIPS 2023).

Answering complex queries over Knowledge Graphs involves a logical reasoning process where a conclusion should be inferred from the available knowledge.

Neural link predictors [Nickel et al., 2016] tackle the problem of identifying missing edges in large KGs. However, in many domains, it is a challenge to develop techniques for answering complex queries involving multiple and potentially unobserved edges, entities, and variables rather than just single edges.

Prior work proposed to address this problem using specialised neural networks trained end-to-end for the query answering task [Hamilton et al., 2018, Daza and Cochez, 2020, Ren et al., 2020, Ren and Leskovec, 2020, Zhu et al., 2022], which offer little interpretability and require training with large and diverse datasets of query-answer pairs. These methods stand in contrast with Complex Query Decomposition [CQD, Arakelyan et al., 2021, Minervini et al., 2022], which showed that it is sufficient to re-use a simple link prediction model to answer complex queries, thus reducing the amount of training data required by orders of magnitude while allowing the possibility to explain intermediate answers. While effective, CQD does not support negations, and fundamentally, it relies on a link predictor whose scores are not necessarily calibrated for the complex query answering task. Adapting a neural link predictor for the query answering task while maintaining the data and parameter efficiency of CQD, as well as its interpretable nature, is the open challenge we take on in this paper.

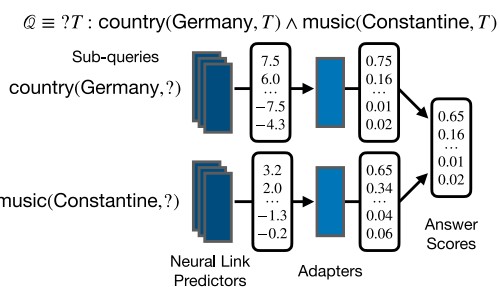

Figure 1: Given a complex query $\mathcal{Q}$, CQD$^{\mathcal{A}}$ adapts the neural link prediction scores for the sub-queries to improve the interactions between them.

We propose CQD$^{\mathcal{A}}$, a lightweight *adaptation* model trained to calibrate link prediction scores, using complex query answering as the optimisation objective. We define the adaptation function as an affine transformation of the original score with a few learnable parameters. The low parameter count and the fact that the adaptation function is independent of the query structure allow us to maintain the efficiency properties of CQD. Besides, the calibration enables a natural extension of CQD to queries with atomic negations.

An evaluation of CQD$^{\mathcal{A}}$ on three benchmark datasets for complex query answering shows an increase from 34.4 to 35.1 MRR over the current state-of-the-art averaged across all datasets while using $\leq 30\%$ of the available training query types. In ablation experiments, we show that the method is data-efficient; it achieves results comparable to the state-of-the-art while using only $1\%$ of the complex queries. Our experiments reveal that CQD$^{\mathcal{A}}$ can generalise across unseen query types while using only $1\%$ of the instances from a single complex query type during training.

## 2 Related Work

**Link Predictors in Knowledge Graphs** Reasoning over KGs with missing nodes has been widely explored throughout the last few years. One can approach the task using latent feature models, such as neural link predictors [Bordes et al., 2013, Trouillon et al., 2016, Yang et al., 2014, Dettmers et al., 2018, Sun et al., 2019, Balažević et al., 2019, Amin et al., 2020] which learn continuous representations for the entities and relation types in the graph and can answer atomic queries over incomplete KGs. Other research lines tackle the link prediction problem through graph feature models [Xiong et al., 2017, Das et al., 2017, Hildebrandt et al., 2020, Yang et al., 2017, Sadeghian et al., 2019], and Graph Neural Networks [GNNs, Schlichtkrull et al., 2018, Vashishth et al., 2019, Teru et al., 2020].

**Complex Query Answering** Complex queries over knowledge graphs can be formalised by extending one-hop atomic queries with First Order Logic (FOL) operators, such as the existential quantifier ($\exists$), conjunctions ($\wedge$), disjunctions ($\vee$) and negations ($\neg$). These FOL constructs can be represented as directed acyclic graphs, which are used by embedding-based methods that represent the queries

using geometric objects [Ren et al., 2020, Hamilton et al., 2018] or probabilistic distributions [Ren and Leskovec, 2020, Zhang et al., 2021, Choudhary et al., 2021] and search the embedding space for the answer set. It is also possible to enhance the properties of the embedding space using GNNs and Fuzzy Logic [Zhu et al., 2022, Chen et al., 2022]. A recent survey [Ren et al., 2023] provides a broad overview of different approaches. Recent work [Daza and Cochez, 2020, Hamilton et al., 2018, Ren and Leskovec, 2020] suggests that such methods require a large dataset with millions of diverse queries during the training, and it can be hard to explain their predictions.

Our work is closely related to CQD [Arakelyan et al., 2021, Minervini et al., 2022], which uses a pre-trained neural link predictor along with fuzzy logical t-norms and t-conorms for complex query answering. A core limitation of CQD is that the pre-trained neural link predictor produces scores not calibrated to interact during the complex query-answering process. This implies that the final scores of the model are highly dependent on the choice of the particular t-(co)norm aggregation functions, which, in turn, leads to discrepancies within the intermediate reasoning process and final predictions. As a side effect, the lack of calibration also means that the equivalent of logical negation in fuzzy logic does not work as expected.

With $\text{CQD}^{\mathcal{A}}$, we propose a solution to these limitations by introducing a scalable adaptation function that calibrates link prediction scores for query answering. Furthermore, we extend the formulation of CQD to support a broader class of FOL queries, such as queries with atomic negation.

## 3 Background

A Knowledge Graph $\mathcal{G} \subseteq \mathcal{E} \times \mathcal{R} \times \mathcal{E}$ can be defined as a set of subject-predicate-object $\langle s, p, o \rangle$ triples, where each triple encodes a relationship of type $p \in \mathcal{R}$ between the subject $s \in \mathcal{E}$ and the object $o \in \mathcal{E}$ of the triple, where $\mathcal{E}$ and $\mathcal{R}$ denote the set of all entities and relation types, respectively. A Knowledge Graph can be represented as a First-Order Logic Knowledge Base, where each triple $\langle s, p, o \rangle$ denotes an atomic formula $p(s, o)$, with $p \in \mathcal{R}$ a binary predicate and $s, o \in \mathcal{E}$ its arguments.

**First-Order Logical Queries** We are concerned with answering logical queries over incomplete knowledge graphs. We consider queries that use existential quantification ($\exists$) and conjunction ($\wedge$) operations. Furthermore, we include disjunctions ($\vee$) and atomic negations ($\neg$). We follow Ren et al. [2020] by transforming a logical query into Disjunctive Normal Form [DNF, Davey and Priestley, 2002], i.e. a disjunction of conjunctive queries, along with the subsequent extension with atomic negations in [Ren and Leskovec, 2020]. We denote such queries as follows:

$$
\begin{aligned}
\mathcal{Q}[A] \triangleq{} &?A : \exists V_1, \ldots, V_m. \left( e_1^1 \wedge \ldots \wedge e_{n_1}^1 \right) \vee \ldots \vee \left( e_1^d \wedge \ldots \wedge e_{n_d}^d \right), \\
&\text{where } e_i^j = p(c, V), \text{ with } V \in \{A, V_1, \ldots, V_m\}, c \in \mathcal{E}, p \in \mathcal{R}, \\
&\text{or } e_i^j = p(V, V'), \text{ with } V, V' \in \{A, V_1, \ldots, V_m\}, V \neq V', p \in \mathcal{R}.
\end{aligned}
\tag{1}
$$

In Equation (1), the variable $A$ is the *target* of the query, $V_1, \ldots, V_m$ denote the *bound variable nodes*, while $c \in \mathcal{E}$ represent the *input anchor nodes*, which correspond to known entities in the query. Each $e_i$ denotes a logical atom, with either one ($p(c, V)$) or two variables ($p(V, V')$).

The goal of answering the logical query $\mathcal{Q}$ consists in finding the answer set $[\![\mathcal{Q}]\!] \subseteq \mathcal{E}$ such that $a \in [\![\mathcal{Q}]\!]$ iff $\mathcal{Q}[a]$ holds true. As illustrated in Figure 1, the *dependency graph* of a conjunctive query $\mathcal{Q}$ is a graph where nodes correspond to variable or non-variable atom arguments in $\mathcal{Q}$ and edges correspond to atom predicates. We follow Hamilton et al. [2018] and focus on queries whose dependency graph is a directed acyclic graph, where anchor entities correspond to source nodes, and the query target $A$ is the unique sink node.

**Example 3.1** (Complex Query). Consider the question "*Which people are German and produced the music for the film Constantine?*". It can be formalised as a complex query $\mathcal{Q} \equiv ?T : \text{country}(\text{Germany}, T) \wedge \text{producerOf}(\text{Constantine}, T)$, where *Germany* and *Constantine* are anchor nodes, and $T$ is the target of the query, as presented in Figure 1. The answer $[\![\mathcal{Q}]\!]$ corresponds to all the entities in the knowledge graph that are German composers for the film Constantine.

**Continuous Query Decomposition** CQD is a framework for answering EPFO logical queries in the presence of missing edges [Arakelyan et al., 2021, Minervini et al., 2022]. Given a query $\mathcal{Q}$, CQD

defines the score of a target node $a \in \mathcal{E}$ as a candidate answer for a query as a function of the score of all atomic queries in $\mathcal{Q}$, given a variable-to-entity substitution for all variables in $\mathcal{Q}$.

Each variable is mapped to an *embedding vector* that can either correspond to an entity $c \in \mathcal{E}$ or to a *virtual entity*. The score of each of the query atoms is determined individually using a neural link predictor [Nickel et al., 2016]. Then, the score of the query with respect to a given candidate answer $\mathcal{Q}[a]$ is computed by aggregating all of the atom scores using t-norms and t-conorms – continuous relaxations of the logical conjunction and disjunction operators.

**Neural Link Predictors**   A neural link predictor is a differentiable model where atom arguments are first mapped into a $d$-dimensional embedding space and then used to produce a score for the atom. More formally, given a query atom $p(s, o)$, where $p \in \mathcal{R}$ and $s, o \in \mathcal{E}$, the score for $p(s, o)$ is computed as $\phi_p(\mathbf{e}_s, \mathbf{e}_o)$, where $\mathbf{e}_s, \mathbf{e}_o \in \mathbb{R}^d$ are the embedding vectors of $s$ and $o$, and $\phi_p : \mathbb{R}^d \times \mathbb{R}^d \mapsto [0, 1]$ is a *scoring function* computing the likelihood that entities $s$ and $o$ are related by the relationship $p$. Following Arakelyan et al. [2021], Minervini et al. [2022], in our experiments, we use a regularised variant of ComplEx [Trouillon et al., 2016, Lacroix et al., 2018] as the neural link predictor of choice, due to its simplicity, efficiency, and generalisation properties [Ruffinelli et al., 2020]. To ensure that the output of the neural link predictor is always in $[0, 1]$, following Arakelyan et al. [2021], Minervini et al. [2022], we use either a sigmoid function or min-max re-scaling.

**T-norms and Negations**   Fuzzy logic generalises over Boolean logic by relaxing the logic conjunction ($\wedge$), disjunction ($\vee$) and negation ($\neg$) operators through the use of t-norms, t-conorms, and fuzzy negations. A *t-norm* $\top : [0, 1] \times [0, 1] \mapsto [0, 1]$ is a generalisation of conjunction in fuzzy logic [Klement et al., 2000, 2004]. Some examples include the *Gödel t-norm* $\top_{\min}(x, y) = \min\{x, y\}$, the *product t-norm* $\top_{\mathrm{prod}}(x, y) = x \times y$, and the *Łukasiewicz t-norm* $\top_{\mathrm{Luk}}(x, y) = \max\{0, x + y - 1\}$.

Analogously, *t-conorms* are dual to t-norms for disjunctions – given a t-norm $\top$, the complementary t-conorm is defined by $\bot(x, y) = 1 - \top(1 - x, 1 - y)$. In our experiments, we use the Gödel t-norm and product t-norm with their corresponding t-conorms.

Fuzzy logic also encompasses negations $n : [0, 1] \mapsto [0, 1]$. The *standard* $n_{\mathrm{stand}}(x) = 1 - x$ and *strict cosine* $n_{\cos} = \frac{1}{2}(1 + \cos(\pi x))$ are common examples of fuzzy negations[Kruse and Moewes, 1993]. To support a broader class of queries, we introduce the *standard* and *strict cosine* functions to model negations in CQD$^{\mathcal{A}}$, which was not considered in the original formulation of CQD.

**Continuous Query Decomposition**   Given a DNF query $\mathcal{Q}$ as defined in Equation (1), CQD aims to find the variable assignments that render $\mathcal{Q}$ true. To achieve this, CQD casts the problem of query answering as an optimisation problem. The aim is to find a mapping from variables to entities $S = \{A \leftarrow a, V_1 \leftarrow v_1, \ldots, V_m \leftarrow v_m\}$, where $a, v_1, \ldots, v_m \in \mathcal{E}$ are entities and $A, V_1, \ldots, V_m$ are variables, that *maximises* the score of $\mathcal{Q}$:

$$
\begin{aligned}
\arg\max_S \mathrm{score}(\mathcal{Q}, S) = \arg\max_{A, V_{1,\ldots,m} \in \mathcal{E}} & \left(e_1^1 \top \ldots \top e_{n_1}^1\right) \bot \ldots \bot \left(e_1^d \top \ldots \top e_{n_d}^d\right) \\
& \text{where } e_i^j = \phi_p(\mathbf{e}_c, \mathbf{e}_V), \text{ with } V \in \{A, V_1, \ldots, V_m\}, c \in \mathcal{E}, p \in \mathcal{R} \\
& \text{or } e_i^j = \phi_p(\mathbf{e}_V, \mathbf{e}_{V'}), \text{ with } V, V' \in \{A, V_1, \ldots, V_m\}, V \neq V', p \in \mathcal{R},
\end{aligned}
\tag{2}
$$

where $\top$ and $\bot$ denote a t-norm and a t-conorm – a continuous generalisation of the logical conjunction and disjunction, respectively – and $\phi_p(\mathbf{e}_s, \mathbf{e}_o) \in [0, 1]$ denotes the neural link prediction score for the atom $p(s, o)$.

**Complex Query Answering via Combinatorial Optimisation**   Following Arakelyan et al. [2021], Minervini et al. [2022], we solve the optimisation problem in Equation (2) by greedily searching for a set of variable substitutions $S = \{A \leftarrow a, V_1 \leftarrow v_1, \ldots, V_m \leftarrow v_m\}$, with $a, v_1, \ldots, v_m \in \mathcal{E}$, that maximises the complex query score, in a procedure akin to *beam search*. We do so by traversing the dependency graph of a query $\mathcal{Q}$ and, whenever we find an atom in the form $p(c, V)$, where $p \in \mathcal{R}$, $c$ is either an entity or a variable for which we already have a substitution, and $V$ is a variable for which we do not have a substitution yet, we replace $V$ with all entities in $\mathcal{E}$ and retain the top-$k$ entities $t \in \mathcal{E}$ that maximise $\phi_p(\mathbf{e}_c, \mathbf{e}_t)$ – i.e. the most likely entities to appear as a substitution of $V$ according to the neural link predictor. As we traverse the dependency graph of a query, we keep a beam with the most promising variable-to-entity substitutions identified so far.

**Example 3.2** (Combinatorial Optimisation). Consider the query "*Which musicians $M$ received awards associated with a genre $g$?*, which can be rewritten as $?M : \exists A.\text{assoc}(g, A) \wedge \text{received}(A, M)$. To answer this query using combinatorial optimisation, we must find the top-$k$ awards $a$ that are candidates to substitute the variable $A$ in $\text{assoc}(g, A)$. This will allow us to understand the awards associated with the genre $g$. Afterwards, for each candidate substitution for $A$, we search for the top-$k$ musicians $m$ that are most likely to substitute $M$ in $\text{received}(A, M)$, ending up with $k^2$ musicians. Finally, we rank the $k^2$ candidates using the final query score produced by a t-norm. ∎

## 4 Calibrating Link Prediction Scores on Complex Queries

The main limitation in the CQD method outlined in Section 3 is that neural link predictors $\phi$ are trained to answer simple, atomic queries, and the resulting answer scores are not trained to interact with one another.

**Example 4.1.** Consider the running example query "*Which people are German and produced the music for the film Constantine?*" which can be rewritten as a complex query $\mathcal{Q} \equiv ?T :$ country(Germany, $T$) $\wedge$ producerOf(Constantine, $T$). To answer this complex query, CQD answers the atomic sub-queries $\mathcal{Q}_1 = \text{country}(\text{Germany}, T)$ and $\mathcal{Q}_2 = \text{producerOf}(\text{Constantine}, T)$ using a neural link predictor, and aggregates the resulting scores using a t-norm. However, the neural link predictor was only trained on answering atomic queries, and the resulting scores are not calibrated to interact with each other. For example, the scores for the atomic queries about the relations country and producerOf may be on different scales, which causes problems when aggregating such scores via t-norms. Let us assume the top candidates for the variable $T$ coming from the atomic queries $\mathcal{Q}_1, \mathcal{Q}_2$ are $\mathcal{A}_1 \leftarrow$ *Sam Shepard* and $\mathcal{A}_2 \leftarrow$ *Klaus Badelt*, with their corresponding neural link prediction scores 1.2 and 8.9, produced using $\phi_{country}$ and $\phi_{producerOf}$. We must also factor in the neural link prediction score of the candidate $\mathcal{A}_1$ for query $\mathcal{Q}_2$ at 7.4 and vice versa at 0.5. When using the Gödel t-norm $\top_{\min}(x, y) = \min\{x, y\}$, the scores associated with the variable assignments $\mathcal{A}_1, \mathcal{A}_2$ are computed as, $\min(8.0, 0.5) = 0.5 \min(7.4, 1.2) = 1.2$. For both answers $\mathcal{A}_1$ and $\mathcal{A}_2$, the scores produced by $\phi_{country}$ for $\mathcal{Q}_1$ are always lower than the scores produced with $\phi_{producerOf}$ for $\mathcal{Q}_2$, meaning that the scores of the latter are not considered when producing the final answer. This phenomenon can be broadly observed in CQD, illustrated in Figure 2. ∎

To address this problem, we propose a method for adaptively learning to calibrate neural link prediction scores by back-propagating through the complex query-answering process. More formally, let $\phi_p$ denote a neural link predictor. We learn an additional adaptation function $\rho_\theta$, parameterised by $\theta = \{\alpha, \beta\}$, with $\alpha, \beta \in \mathbb{R}$. Then, we use the composition of $\rho_\theta$ and $\phi_p$, $\rho_\theta \circ \phi_p$, such that:

$$\rho_\theta(\phi_p(\mathbf{e}_V, \mathbf{e}_{V'})) = \phi_p(\mathbf{e}_V, \mathbf{e}_{V'})(1 + \alpha) + \beta. \tag{3}$$

Here, the function $\rho$ defines an affine transformation of the score and when the parameters $\alpha = \beta = 0$, the transformed score $\rho_\theta(\phi_p(\mathbf{e}_V, \mathbf{e}_{V'}))$ recovers the original scoring function. The parameters $\theta$ can be conditioned on the representation of the predicate $p$ and the entities $V$ and $V'$, i.e. $\theta = \psi(\mathbf{e}_V, \mathbf{e}_p, \mathbf{e}_{V'})$; here, $\psi$ is an end-to-end differentiable neural module with parameters $\mathbf{W}$. $\mathbf{e}_V$, $\mathbf{e}_p$, $\mathbf{e}_{V'}$ respectively denote the representations of the subject, predicate, and object of the atomic query. In our experiments, we consider using one or two linear transformation layers with a ReLU non-linearity as options for $\psi$.

The motivation for our proposed adaptation function is twofold. Initially, it is monotonic, which is desirable for maintaining the capability to interpret intermediate scores, as in the original formulation of CQD. Moreover, we draw inspiration from the use of affine transformations in methodologies such as Platt scaling [Platt et al., 1999], which also use a linear function for calibrating probabilities and have been applied in the problem of calibration of link prediction models [Tabacof and Costabello, 2020]. Parameter-efficient adaptation functions have also been applied effectively in other domains, such as adapter layers Houlsby et al. [2019] used for fine-tuning language models in NLP tasks.

**Training** For training the score calibration component in Equation (3), we first compute how likely each entity $a' \in \mathcal{E}$ is to be an answer to the query $\mathcal{Q}$. To this end, for each candidate answer $a' \in \mathcal{E}$, we compute the *answer score* as the complex query score assuming that $a' \in \mathcal{E}$ is the final answer as:

$$\text{score}(\mathcal{Q}, A \leftarrow a') = \max_S \text{score}(\mathcal{Q}, S), \text{ where } A \leftarrow a' \in S. \tag{4}$$

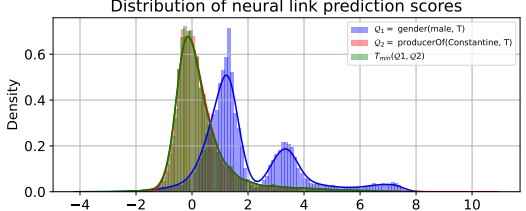

Figure 2: The distributions of two atomic scores $\mathcal{Q}_1$ and $\mathcal{Q}_2$, and the aggregated results via $\top_{\min}$ – the scores from $\mathcal{Q}_2$ dominate the final scores.

| Split | Query Types | FB15K | FB15K-237 | NELL995 |
|-------|-------------|-------|-----------|---------|
| **Train** | 1p, 2p, 3p, 2i, 3i | 273,710 | 149,689 | 107,982 |
|  | 2in, 3in, inp, pin, pni | 27,371 | 14,968 | 10,798 |
| **Valid** | 1p | 59,078 | 20,094 | 16,910 |
|  | Others | 8,000 | 5,000 | 4,000 |
| **Test** | 1p | 66,990 | 22,804 | 17,021 |
|  | Others | 8,000 | 5,000 | 4,000 |

Table 1: Statistics on the different types of query structures in FB15K, FB15K-237, and NELL995.

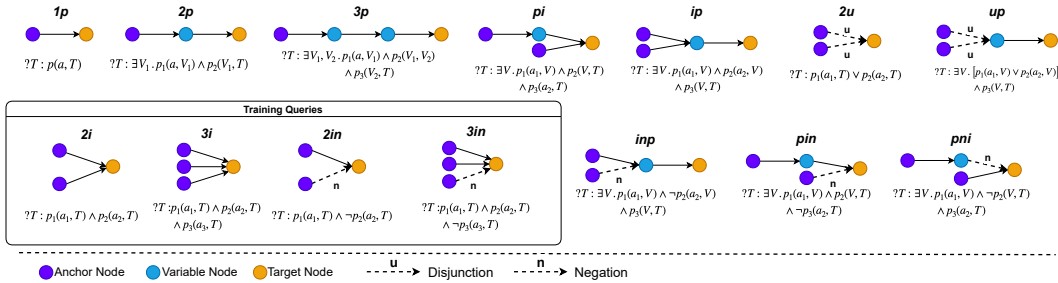

Figure 3: Query structures considered in our experiments, as proposed by Ren and Leskovec [2020] – the naming of each query structure corresponds to *projection* (**p**), *intersection* (**i**), *union* (**u**) and *negation* (**n**), reflecting how they were generated in the BetaE paper [Ren and Leskovec, 2020]. An example of a **pin** query is $?T : \exists V.p(a, V), q(V, T), \neg r(b, T)$, where $a$ and $b$ are anchor nodes, $V$ is a variable node, and $T$ is the query target node.

Equation (4) identifies the variable-to-entity substitution $S$ that 1) maximises the query score score$(\mathcal{Q}, S)$, defined in Equation (2), and 2) associates the answer variable $A$ with $a' \in \mathcal{E}$, i.e. $A \leftarrow a' \in S$. For computing $S$ with the additional constraint that $A \leftarrow a' \in S$, we use the complex query answering procedure outlined in Section 3. We optimise the additional parameters $\mathbf{W}$ introduced in Section 4, by gradient descent on the likelihood of the true answers on a dataset $\mathcal{D} = \{(\mathcal{Q}_i, a_i)\}_{i=1}^{|\mathcal{D}|}$ of query-answer pairs by using a *1-vs-all* cross-entropy loss, introduced by Lacroix et al. [2018], which was also used to train the neural link prediction model:

$$\mathcal{L}(\mathcal{D}) = \sum_{(\mathcal{Q}_i, a_i) \in \mathcal{D}} -\text{score}(\mathcal{Q}_i, A \leftarrow a_i) + \log \left[ \sum_{a' \in \mathcal{E}} \exp\left(\text{score}(\mathcal{Q}_i, A \leftarrow a')\right) \right]. \quad (5)$$

In addition to the *1-vs-all* [Ruffinelli et al., 2020] loss in Equation (5), we also experiment with the binary cross-entropy loss, using the negative sampling procedure from Ren and Leskovec [2020].

## 5 Experiments

**Datasets** To evaluate the complex query answering capabilities of our method, we use a benchmark comprising of 3 KGs: FB15K [Bordes et al., 2013], FB15K-237 [Toutanova and Chen, 2015] and NELL995 [Xiong et al., 2017]. For a fair comparison with previous work, we use the datasets of FOL queries proposed by Ren and Leskovec [2020], which includes nine structures of EPFO queries and 5 query types with atomic negations, seen in Figure 3. The datasets provided by Ren and Leskovec [2020] introduce queries with *hard* answers, which are the answers that cannot be obtained by direct graph traversal; in addition, this dataset does not include queries with more than 100 answers, increasing the difficulty of the complex query answering task. The statistics for each dataset can be seen in Table 1. Note that during training, we only use *2i*, *3i*, *2in*, and *3in* queries, corresponding to $\leq 30\%$ of the training dataset, for the adaptation of the neural link predictor. To assess the model's ability to generalise, we evaluate it on all query types.

| Model | avg$_p$ | avg$_n$ | 1p | 2p | 3p | 2i | 3i | pi | ip | 2u | up | 2in | 3in | inp | pin | pni |
|---|---|---|---|---|---|---|---|---|---|---|---|---|---|---|---|---|
| **FB15K** | | | | | | | | | | | | | | | | |
| GQE | 28.0 | - | 54.6 | 15.3 | 10.8 | 39.7 | 51.4 | 27.6 | 19.1 | 22.1 | 11.6 | - | - | - | - | - |
| Q2B | 38.0 | - | 68.0 | 21.0 | 14.2 | 55.1 | 66.5 | 39.4 | 26.1 | 35.1 | 16.7 | - | - | - | - | - |
| BetaE | 41.6 | 11.8 | 65.1 | 25.7 | 24.7 | 55.8 | 66.5 | 43.9 | 28.1 | 40.1 | 25.2 | 14.3 | 14.7 | 11.5 | 6.5 | 12.4 |
| CQD-CO | 46.9 | - | **89.2** | 25.3 | 13.4 | 74.4 | 78.3 | 44.1 | 33.2 | 41.8 | 21.9 | - | - | - | - | - |
| CQD-Beam | 68.4 | - | **89.2** | 65.3 | 29.7 | 76.1 | 79.3 | 70.6 | 70.6 | 72.3 | 59.4 | - | - | - | - | - |
| ConE | 49.8 | 14.8 | 73.3 | 33.8 | 29.2 | 64.4 | 73.7 | 50.9 | 35.7 | 55.7 | 31.4 | 17.9 | 18.7 | 12.5 | 9.8 | 15.1 |
| GNN-QE | **72.8** | 38.6 | 88.5 | **69.3** | **58.7** | **79.7** | **83.5** | 69.9 | 70.4 | **74.1** | **61.0** | 44.7 | 41.7 | **42.0** | 30.1 | **34.3** |
| CQD$^{\mathcal{A}}$ | 70.4 | **42.8** | **89.2** | 64.5 | 57.9 | 76.1 | 79.4 | **70.0** | 70.6 | 68.4 | 57.9 | **54.7** | **47.1** | 37.6 | **35.3** | 24.6 |
| **FB15K-237** | | | | | | | | | | | | | | | | |
| GQE | 16.3 | - | 35.0 | 7.2 | 5.3 | 23.3 | 34.6 | 16.5 | 10.7 | 8.2 | 5.7 | - | - | - | - | - |
| Q2B | 20.1 | - | 40.6 | 9.4 | 6.8 | 29.5 | 42.3 | 21.2 | 12.6 | 11.3 | 7.6 | - | - | - | - | - |
| BetaE | 20.9 | 5.5 | 39.0 | 10.9 | 10.0 | 28.8 | 42.5 | 22.4 | 12.6 | 12.4 | 9.7 | 5.1 | 7.9 | 7.4 | 3.5 | 3.4 |
| CQD-CO | 21.8 | - | **46.7** | 9.5 | 6.3 | 31.2 | 40.6 | 23.6 | 16.0 | 14.5 | 8.2 | - | - | - | - | - |
| CQD-Beam | 25.3 | - | **46.7** | 13.3 | 7.9 | 34.4 | 48.3 | 27.1 | 20.4 | 17.6 | 11.5 | - | - | - | - | - |
| ConE | 23.4 | 5.9 | 41.8 | 12.8 | 11.0 | 32.6 | 47.3 | 25.5 | 14.0 | 14.5 | 10.8 | 5.4 | 8.6 | 7.8 | 4.0 | 3.6 |
| GNN-QE | **26.8** | 10.2 | 42.8 | **14.7** | **11.8** | **38.3** | **54.1** | **31.1** | 18.9 | 16.2 | **13.4** | 10.0 | **16.8** | **9.3** | 7.2 | **7.8** |
| CQD$^{\mathcal{A}}$ | 25.7 | **10.7** | **46.7** | 13.6 | 11.4 | 34.5 | 48.3 | 27.4 | **20.9** | **17.6** | 11.4 | **13.6** | **16.8** | 7.9 | **8.9** | 5.8 |
| **NELL995** | | | | | | | | | | | | | | | | |
| GQE | 18.6 | - | 32.8 | 11.9 | 9.6 | 27.5 | 35.2 | 18.4 | 14.4 | 8.5 | 8.8 | - | - | - | - | - |
| Q2B | 22.9 | - | 42.2 | 14.0 | 11.2 | 33.3 | 44.5 | 22.4 | 16.8 | 11.3 | 10.3 | - | - | - | - | - |
| BetaE | 24.6 | 5.9 | 53.0 | 13.0 | 11.4 | 37.6 | 47.5 | 24.1 | 14.3 | 12.2 | 8.5 | 5.1 | 7.8 | 10.0 | 3.1 | 3.5 |
| CQD-CO | 28.8 | - | **60.4** | 17.8 | 12.7 | 39.3 | 46.6 | 30.1 | 22.0 | 17.3 | 13.2 | - | - | - | - | - |
| CQD-Beam | 31.8 | - | **60.4** | 22.6 | 13.6 | 42.6 | 52.0 | 31.2 | 25.6 | 19.9 | 16.7 | - | - | - | - | - |
| ConE | 27.2 | 6.4 | 53.1 | 16.1 | 13.9 | 40.0 | 50.8 | 26.3 | 17.5 | 15.3 | 11.3 | 5.7 | 8.1 | 10.8 | 3.5 | 3.9 |
| GNN-QE | 28.9 | 9.7 | 53.3 | 18.9 | 14.9 | 42.4 | 52.5 | 30.8 | 18.9 | 15.9 | 12.6 | 9.9 | 14.6 | 11.4 | 6.3 | 6.3 |
| CQD$^{\mathcal{A}}$ | **32.3** | **13.3** | **60.4** | **22.9** | **16.7** | **43.4** | **52.6** | **32.1** | **26.4** | **20.0** | **17.0** | **15.1** | **18.6** | **15.8** | **10.7** | **6.5** |

Table 2: MRR results for FOL queries on the testing sets. **avg**$_p$ designates the averaged results for EPFO queries ($\wedge$, $\vee$), while **avg**$_n$ pertains to queries including atomic negations ($\neg$). The results for the baselines are from Zhu et al. [2022].

**Evaluation Protocol**     For a fair comparison with prior work, we follow the evaluation scheme in Ren and Leskovec [2020] by separating the answer of each query into *easy* and *hard* sets. For test and validation splits, we define *hard* queries as those that cannot be answered via direct traversal along the edges of the KG and can only be answered by predicting at least one missing link, meaning *non-trivial* reasoning should be completed. We evaluate the method on non-trivial queries by calculating the rank $r$ for each hard answer against non-answers and computing the Mean Reciprocal Rank (MRR).

**Baselines**     We compare CQD$^{\mathcal{A}}$ with state-of-the-art methods from various solution families in Section 2. In particular, we choose GQE [Hamilton et al., 2018], Query2Box [Ren et al., 2020], BetaE [Ren and Leskovec, 2020] and ConE [Zhang et al., 2021] as strong baselines for query embedding methods. We also compare with methods based on GNNs and fuzzy logic, such as FuzzQE [Chen et al., 2022], GNN-QE [Zhu et al., 2022], and the original CQD [Arakelyan et al., 2021, Minervini et al., 2022], which uses neural link predictors for answering EPFO queries without any fine-tuning on complex queries.

**Model Details**     Our method can be used with any neural link prediction model. Following Arakelyan et al. [2021], Minervini et al. [2022], we use ComplEx-N3 [Lacroix et al., 2018]. We identify the optimal hyper-parameters using the validation MRR. We train for $50,000$ steps using Adagrad as an optimiser and $0.1$ as the learning rate. The beam-size hyper-parameter $k$ was selected in $k \in \{512, 1024, \ldots, 8192\}$, and the loss was selected across *1-vs-all* [Lacroix et al., 2018] and binary cross-entropy with one negative sample.

**Parameter Efficiency**     We use the query types *2i*, *3i*, *2in*, *3in* for training the calibration module proposed in Section 4. We selected these query types as they do not require variable assignments other than for the answer variable $A$, making the training process efficient. As the neural link prediction model is frozen, we only train the adapter layers that have a maximum of $\mathbf{W} \in \mathbb{R}^{2 \times 2d}$ learnable weights. Compared to previous works, we have $\approx 10^3$ times fewer *trainable* parameters, as shown in Table 3, while maintaining competitive results.

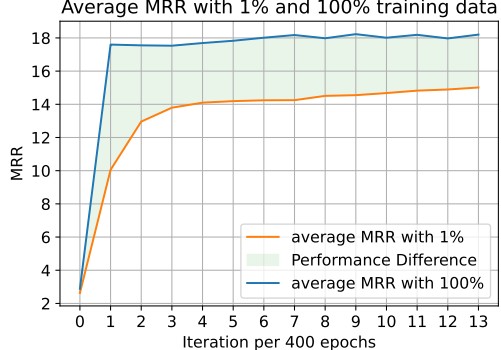

**Average MRR with 1% and 100% training data**

| | legend |
|---|---|
| | average MRR with 1% |
| | Performance Difference |
| | average MRR with 100% |

Figure 4: Average test MRR score ($y$-axis) of CQD$^{\mathcal{A}}$ using $1\%$ and $100\%$ of the training queries from FB15K-237 throughout the training iterations ($x$-axis).

| | Number of parameters | | |
|---|---|---|---|
| | FB15K | FB15K-237 | NELL |
| CQD$^{\mathcal{A}}$ | $1.3 \times 10^7$ 
 frozen 
 $\mathbf{+4 \times 10^3}$ | $1.3 \times 10^7$ 
 frozen 
 $\mathbf{+4 \times 10^3}$ | $7.5 \times 10^7$ 
 frozen 
 $\mathbf{+4 \times 10^3}$ |
| BetaE | $1.3 \times 10^7$ | $1.3 \times 10^7$ | $6 \times 10^7$ |
| Q2B | $1.2 \times 10^7$ | $1.2 \times 10^7$ | $6 \times 10^7$ |
| GNN-QE | $3 \times 10^6$ | $3 \times 10^6$ | $3 \times 10^6$ |
| ConE | $1.2 \times 10^7$ | $1.2 \times 10^7$ | $6 \times 10^7$ |
| GQE | $1.5 \times 10^7$ | $1.5 \times 10^7$ | $7.5 \times 10^7$ |

Table 3: Number of parameters used by different complex query answering methods – values for GNN-QE are approximated using the backbone NBFNet [Zhu et al., 2021], while the remaining use their original studies.

## 5.1 Results

**Complex Query Answering**   Table 2 shows the predictive accuracy of CQD$^{\mathcal{A}}$ for answering complex queries compared to the current state-of-the-art methods. Some methods do not support queries that include negations; we leave the corresponding entries blank. We can see that CQD$^{\mathcal{A}}$ increases the MRR from $34.4$ to $35.1$ averaged across all query types and datasets. In particular, CQD$^{\mathcal{A}}$ shows the most substantial increase in predictive accuracy on NELL995 by producing more accurate results than all other methods for all query types.   CQD$^{\mathcal{A}}$ can achieve these results using $\leq 30\%$ of the complex query types during training while maintaining competitive results across each dataset and query type. For queries including negations, CQD$^{\mathcal{A}}$ achieves a relative improvement of $6.8\%$ to $37.1\%$, which can be attributed to the fact that the adaptation is completed with query types *2in* and *3in* that include negation, which allows for learning an adaptation layer that is robust for these types of queries. In our experiments, we found that calculating the neural adaptation parameters $\theta$ of the adaptation function $\rho_\theta$ in Equation (3) as a function of the predicate representation yields the most accurate results followed by computing $\theta$ as a function of the source entity and predicate representation, which is strictly more expressive. In Appendix A, we show the impact of the adaptation layers on the neural link prediction scores.

The adaptation process does not require data-intensive training and allows the model to generalise to query types not observed during training. This prompts us to investigate the minimal amount of data samples and query types required for adaptation.

**Data Efficiency**   To analyse the data efficiency of CQD$^{\mathcal{A}}$, we compare the behaviour of the pre-trained link predictors tuned with $1\%$ and $100\%$ of the training complex query examples in FB15K-237, presented in Table 4. For adapting on $1\%$ of the training complex queries, we used the same hyper-parameters we identified when training on the full dataset. Even when using $1\%$ of the complex training queries (3290 samples) for tuning, the model still achieves competitive results, with an average MRR difference of $2.2$ compared to the model trained using the entire training set. CQD$^{\mathcal{A}}$ also produces higher test MRR results than GNN-QE with an average MRR increase of $4.05$.

We can also confirm that the adaptation process converges after $\leq 10\%$ of the training epochs as seen in Figure 4. The convergence rate is not hindered when using only $1\%$ of the training queries. This shows that CQD$^{\mathcal{A}}$ is a scalable method with a fast convergence rate that can be trained in a data-efficient manner.

**Out-of-Distribution Generalisation**   To study the generalisation properties of CQD$^{\mathcal{A}}$, we trained the adaptation layer on all atomic queries and only $1\%$ of samples for *one* training query type *2i*, one of the simplest complex query types. We see in Table 4 that CQD$^{\mathcal{A}}$ can generalise to other types of complex queries not observed during training with an average MRR difference of $2.9$ compared to training on all training query types. CQD$^{\mathcal{A}}$ also produces significantly higher test MRR results

| Dataset | Model | 1p | 2p | 3p | 2i | 3i | pi | ip | 2u | up | 2in | 3in | inp | pin | pni |
|---|---|---|---|---|---|---|---|---|---|---|---|---|---|---|---|
| | CQD$^{\mathcal{A}}$ | **46.7** | **11.8** | **11.4** | **33.6** | 41.2 | **24.82** | **17.81** | **16.45** | 8.74 | 10.8 | **13.86** | 5.93 | **5.38** | **14.82** |
| FB237, 1% | GNN-QE | 36.82 | 8.96 | 8.13 | 33.02 | **49.28** | 24.58 | 14.18 | 10.73 | 8.47 | 4.89 | 12.31 | **6.74** | 4.41 | 4.09 |
| | BetaE | 36.80 | 6.89 | 5.94 | 22.84 | 34.34 | 17.12 | 8.72 | 9.23 | 5.66 | 4.44 | 6.14 | 5.18 | 2.54 | 2.94 |
| | CQD$^{\mathcal{A}}$ | **46.7** | **11.8** | **11.2** | 30.35 | 40.75 | **23.36** | 18.28 | 15.85 | 8.96 | 9.36 | 10.25 | 5.17 | 4.46 | **4.44** |
| FB237 2i, 1% | GNN-QE | 34.81 | 5.40 | 5.17 | 30.12 | **48.88** | 23.06 | 12.65 | 9.85 | 5.26 | 4.26 | 12.5 | 4.43 | 0.71 | 1.98 |
| | BetaE | 37.99 | 5.62 | 4.48 | 23.73 | 35.25 | 15.63 | 7.96 | 9.73 | 4.56 | 0.15 | 0.49 | 0.62 | 0.10 | 0.14 |

Table 4: Comparison of test MRR results for queries on FB15K-237 using the following training sets – FB237, 1% (resp. FB237 2i, 1%) means that, in addition to all 1p (atomic) queries, only 1% of the complex queries (resp. *2i* queries) was used during training. As CQD$^{\mathcal{A}}$ uses a pre-trained link predictor, we also include all *1p* queries when training GNN-QE for a fair comparison.

| Model | 2p | 2i | 3i | pi | ip | 2u | up | 2in | 3in | inp | pin | pni |
|---|---|---|---|---|---|---|---|---|---|---|---|---|
| CQD | **13.2** | 34.5 | 48.2 | 26.8 | 20.3 | 17.4 | 10.3 | 5.4 | 12.4 | 6.1 | 3.2 | 4.6 |
| CQD$_F$ | 9.3 | 22.8 | 34.9 | 19.8 | 14.5 | 13.0 | 7.2 | 7.4 | 7.1 | 4.9 | 3.9 | 3.8 |
| CQD$_F^{\mathcal{A}}$ | 9.5 | 23.9 | 39.0 | 19.8 | 14.5 | 14.2 | 7.2 | 8.4 | 9.7 | 4.9 | 4.2 | 3.6 |
| CQD$_C$ | 10.9 | 33.7 | 47.3 | 25.6 | 18.9 | 16.4 | 9.4 | 7.9 | 12.2 | 6.6 | 4.2 | 5.0 |
| CQD$_R$ | 6.4 | 22.2 | 31.0 | 16.6 | 11.2 | 12.5 | 4.8 | 4.7 | 5.9 | 4.1 | 2.0 | 3.5 |
| CQD$^{\mathcal{A}}$ | **13.2** | **35.0** | **48.5** | **27.3** | **20.7** | **17.6** | **10.5** | **13.2** | **14.9** | **7.4** | **7.8** | **5.5** |

Table 5: Test MRR results for FOL queries on FB15K-237 using the following CQD extensions: CQD from Arakelyan et al. [2021], Minervini et al. [2022] with the considered normalisation and negations; CQD$_F$, where we fine-tune all neural link predictor parameters in CQD; CQD$_F^{\mathcal{A}}$, where we *fine-tune all link predictor parameters* in CQD$^{\mathcal{A}}$; CQD$_R$, where we learn a *transformation* for the entity and relation embeddings and we use it to *replace* the initial entity and relation representations; and CQD$_C$, where we learn a transformation for the entity and relation embeddings, and we *concatenate* it to the initial entity and relation representations.

than GNN-QE, with an average increase of 5.1 MRR. The greatest degradation in predictive accuracy occurs for the queries containing negations, with an average decrease of 2.7. This prompts us to conjecture that being able to answer general EPFO queries is not enough to generalise to the larger set of queries, which include atomic negation. However, our method can generalise on all query types, using only 1% of the *2i* queries, with 1496 overall samples for adaptation.

**Fine-Tuning All Model Parameters**    One of the reasons for the efficiency of CQD$^{\mathcal{A}}$ is that the neural link predictor is not fine-tuned for query answering, and only the parameters in the adaptation function are learned. We study the effect of fine-tuning the link predictor using the full training data for CQD and CQD$^{\mathcal{A}}$ on FB15K-237. We consider several variants: 1) CQD$_F$, where we **F**ine-tune all neural link predictor parameters in CQD; 2) CQD$_F^{\mathcal{A}}$, where we fine-tune all link predictor parameters in CQD$^{\mathcal{A}}$, 3) CQD$_R$, where we learn a transformation for the entity and relation embeddings and we use it to **R**eplace the initial entity and relation representations, and 4) CQD$_C$, where we learn a transformation for the entity and relation embeddings, and we **C**oncatenate it to the initial entity and relation representations.

It can be seen from Table 5 that CQD$^{\mathcal{A}}$ yields the highest test MRR results across all query types while fine-tuning all the model parameters produces significant degradation along all query types, which we believe is due to catastrophic forgetting [Goodfellow et al., 2013] of the pre-trained link predictor.

# 6   Conclusions

In this work, we propose the novel method CQD$^{\mathcal{A}}$ for answering complex FOL queries over KGs, which increases the averaged MRR over the previous state-of-the-art from 34.4 to 35.1 while using $\leq 30\%$ of query types. Our method uses a single adaptation layer over neural link predictors, which allows for training in a data-efficient manner. We show that the method can maintain competitive predictive accuracy even when using 1% of the training data. Furthermore, our experiments on training on a subset (1%) of the training queries from a single query type (*2i*) show that it can generalise to

new queries that were not used during training while being data-efficient. Our results provide further evidence for how neural link predictors exhibit a form of compositionality that generalises to the complex structures encountered in the more general problem of query answering. $CQD^A$ is a method for improving this compositionality while preserving computational efficiency. As a consequence, rather than designing specialised models trained end-to-end for the query answering task, we can focus our efforts on improving the representations learned by neural link predictors, which would then transfer to query answering via efficient adaptation, as well as other downstream tasks where they have already proved beneficial, such as clustering, entity classification, and information retrieval.

### Acknowledgements

Pasquale was partially funded by the European Union's Horizon 2020 research and innovation programme under grant agreement no. 875160, ELIAI (The Edinburgh Laboratory for Integrated Artificial Intelligence) EPSRC (grant no. EP/W002876/1), an industry grant from Cisco, and a donation from Accenture LLP, and is grateful to NVIDIA for the GPU donations. Daniel and Michael were partially funded by Elsevier's Discovery Lab. Michael was partially funded by the Graph-Massivizer project (Horizon Europe research and innovation program of the European Union under grant agreement 101093202). Erik is partially funded by a DFF Sapere Aude research leader grant under grant agreement No 0171-00034B, as well as by a NEC PhD fellowship, and is supported by the Pioneer Centre for AI, DNRF grant number P1. Isabelle is partially funded by a DFF Sapere Aude research leader grant under grant agreement No 0171-00034B, as well as by the Pioneer Centre for AI, DNRF grant number P1. This work was supported by the Edinburgh International Data Facility (EIDF) and the Data-Driven Innovation Programme at the University of Edinburgh.

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

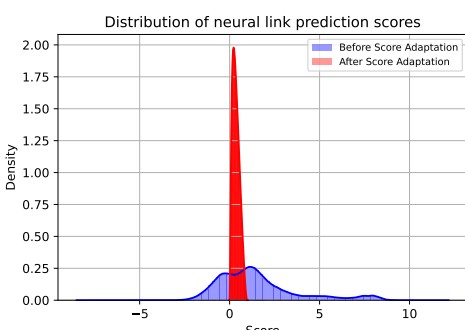

Figure 5: The distribution of the scores of the neural link predictor before applying the adaptation layer and after.

Figure 6: Evaluation of CQD$^{\mathcal{A}}$ using $1\%$ and $100\%$ of the training complex queries during tuning (top) and **2i** queries (bottom) from FB15K-237.

## A  Impact of adaptation

We investigate the effect of the adaptation process in CQD$^{\mathcal{A}}$ by comparing the score of the neural link predictor before and after applying the adaptation layer. As we see from Figure 5, the scores before adaptation have a variation of $5.04$ with the boundaries at $[-8, 12]$. This makes them problematic for complex query answering as discussed in Section 4. The Adapted scores have a smaller variation at $0.03$ while the maximum and minimum lie in the $[0, 1]$ range.

## B  On data efficiency and generalisation of CQD$^{\mathcal{A}}$

We conduct a series of experiments comparing the performance of CQD$^{\mathcal{A}}$ trained while only using $1\%$ of the training data to the complete training set in FB15K-237. We see from Figure 6 (top) that the model maintains a strong performance on the complex reasoning task with a $2.4$ averaged MRR degradation compared to using the complete data. This phenomenon is even more pronounced when we use only $1\%$ of the queries while using *2i* as our training query types Figure 6 (bottom). In this data and query type constrained mode CQD$^{\mathcal{A}}$ maintains competitive performance with a degradation of 2.7 averaged MRR for complex queries compared to training with complete data. This highlights the data-efficient nature of CQD$^{\mathcal{A}}$ and shows that training a score adaptation layer does not require significant data for training. This is also accompanied by the observation that CQD$^{\mathcal{A}}$ is able to generalise to unseen query types while training only on 2i. This behaviour is not inherent, as seen from Table 4.

