# OpenReview forum: "Adapting Neural Link Predictors for Data-Efficient Complex Query Answering"
_NeurIPS.cc/2023/Conference — NeurIPS 2023 poster_

### Official Review · Reviewer_5Xze · 2023-07-05

**Soundness:** 3 good
**Presentation:** 3 good
**Contribution:** 3 good
**Rating:** 7
**Confidence:** 1

**Summary:**

This paper addresses the challenging task of answering complex queries on incomplete knowledge graphs, where missing knowledge introduces additional complexity. Previous approaches either employed end-to-end architectures with opaque reasoning processes or relied on simple neural link predictors, sacrificing information gain for computational efficiency. To overcome these limitations, the authors propose a parameter-efficient score adaptation model optimized for recalibrating neural link prediction scores in complex query answering. While the neural link predictor remains frozen, the adaptation component, with minimal additional parameters, is trained on the downstream task. The proposed method significantly improves accuracy compared to state-of-the-art approaches, achieving higher Mean Reciprocal Rank values while using only a fraction of the available training query types.

**Strengths:**

### Originality
The paper introduces a novel approach to addressing the task of answering complex queries on incomplete knowledge graphs. The use of a parameter-efficient score adaptation model, optimized for recalibrating neural link prediction scores, sets it apart from previous methods. This original approach demonstrates innovation in the field of complex query answering and contributes to advancing the understanding of addressing missing knowledge in knowledge graphs.

### Quality
The paper provides thorough evaluations and comparisons with state-of-the-art methods, demonstrating that the proposed approach produces significantly more accurate results. The proposed method significantly improves accuracy compared to state-of-the-art approaches, achieving higher Mean Reciprocal Rank values while using only a fraction of the available training query types.

### Clarity
The paper effectively communicates the proposed approach and its components, such as the parameter-efficient score adaptation model and the frozen neural link predictor. The writing is clear, concise, and well-structured.

### Significance
The paper addresses an important and challenging task in the field of complex query answering on incomplete knowledge graphs. By introducing the approach and demonstrating its superior performance compared to existing methods, the paper makes a significant contribution to the advancement of techniques for handling missing knowledge in knowledge graphs.


**Weaknesses:**

The authors haven't discussed the limitation of this work in the paper.

**Questions:**

N/A

**Limitations:**

This is a solid work.

---

> ### Author Rebuttal · Authors · 2023-08-08
>
> Thank you for your review. Regarding the following,
> - The authors haven't discussed the limitation of this work in the paper.
>
> We tried our best to run all possible ablations and analyses for this paper, and some additional experiments to run came up during this rebuttal phase. We will add results of CQD with the considered negation functions to the camera ready and a more detailed analysis of the explainability properties of CQD$^\mathcal{A}$ -- which we assume are inherited from CQD, but we do not discuss it in the main paper mainly for space reasons; as well as potential future work directions.

---

> > ### Comment · Reviewer_5Xze · 2023-08-20
> >
> > Thanks for the clarifications!

---

### Official Review · Reviewer_pnJH · 2023-07-06

**Soundness:** 3 good
**Presentation:** 3 good
**Contribution:** 3 good
**Rating:** 7
**Confidence:** 5

**Summary:**

The paper proposes a score adaptation model called CQDA for efficient complex query answering on incomplete knowledge graphs. The authors address the problems in existing methods that are either hard to interpret or require intensive training. CQDA is a parameter-efficient model that recalibrates neural link prediction scores for the complex query answering task. The authors demonstrate that CQDA outperforms state-of-the-art methods in terms of accuracy, data efficiency, and robustness.

**Strengths:**

- The paper addresses an important problem in the field of complex query answering on incomplete knowledge graphs. The proposed CQDA model provides a solution that improves accuracy while maintaining computational efficiency.
- The paper introduces a parameter-efficient score adaptation model that recalibrates neural link prediction scores. This approach reduces the need for extensive training data and resources.
- Experimental results show that CQDA achieves significantly better results than current state-of-the-art methods. The model is also shown to be data-efficient and robust in out-of-domain evaluations.


**Weaknesses:**

I think there is a discrepancy between Figure 1 and the statement, and thus the paper lacks an important baseline: **normalizing scores then merging**.

In Figure 1, the paper shows that the score scale may be different for different subqueries, which makes me think that the original COD may be without normalization which maps the score from any scale to [0,1]. However, the paper also mentions that they follow previous work to add normalization as below:
> To ensure that the output of the neural link predictor is always in [0, 1], following Arakelyan et al. [2021], Minervini et al. [2022], we use either a sigmoid function or min-max re-scaling.

 So, what would happen if we do a merge after normalization on scores for all subqueries? Please correct me if I miss anything.

**Questions:**

- Figure 1 shows the scores over all entities but does not clearly present if the score is from a score function, or from a linear layer. If I understand correctly, the score before adaptation should come from the score function $\phi$, right?

---

> ### Author Rebuttal · Authors · 2023-08-08
>
> Thank you for your review! Regarding your concerns –
> - In Figure 1, the paper shows that the score scale may be different for different subqueries, [..] However, the paper also mentions that they follow previous work to add normalization as below [..]
>
> In CQD, for the t-norm and t-conorms to be applicable, link prediction scores need to be mapped to the $[0, 1]$ range, which they achieve, for example, via the sigmoid function. However, we find that these transformations may be insufficient to solve the score calibration problem outlined in Figure 1. For instance, given two atoms $a_1$ and $a_2$ with scores in $[-10, -5]$ and $[5, 10]$, using the sigmoid will cause the score of the former atom to be close to $0$ and the score of the latter atom to be close to $1$, which will cause the latter score always to be ignored when applying the minimum t-norm. In this work, we propose a solution to this problem by learning to adapt the scores using simple transformation learned on the downstream complex query answering task by back-propagating through the complex query answering process. In our rebuttal PDF (Table 1), we included an additional experiment that evaluates previous work with normalization but without calibration, which we observe produces less accurate results than our proposed CQD$^{\mathcal{A}}$.

---

> > ### Comment · Reviewer_pnJH · 2023-08-18
> > **Response to the Rebuttal**
> >
> > Thanks for the author for the clarification. It makes me more confident that this paper should be accepted.

---

### Official Review · Reviewer_jfF5 · 2023-07-06

**Soundness:** 4 excellent
**Presentation:** 4 excellent
**Contribution:** 4 excellent
**Rating:** 7
**Confidence:** 4

**Summary:**

The paper proposes a novel approach called CQD^A for answering complex queries on incomplete knowledge graphs. The authors address the challenge of answering complex logical queries in the presence of missing knowledge by re-calibrating neural link prediction scores. They introduce an adaptation component that is trained on the downstream complex query answering task, while the neural link predictor is frozen. The proposed model, CQD^A, significantly improves the Mean Reciprocal Rank values compared to existing state-of-the-art methods. It achieves this by increasing the accuracy of answers while using a smaller amount of training data. Additionally, CQD^A is shown to be data-efficient and robust in out-of-domain evaluations.

To sum up, the proposed method is simple yet efficient. I have no reason to reject.

**Strengths:**

1. Novel Approach: The paper introduces a unique and novel approach for answering complex queries on incomplete knowledge graphs. The use of neural link predictors and the adaptation component allows for efficient complex query answering while providing interpretable answers. Well motivation.
2. Improved Accuracy: The proposed model, CQD^A, outperforms current state-of-the-art methods. This improvement demonstrates the effectiveness of the re-calibrated neural link prediction scores in producing more accurate results.
3. Data Efficiency: CQD^A achieves competitive results even with only 1% of the training data. This data efficiency is advantageous as it reduces the computational and resource requirements for training the model.
4. Robustness: The model's robustness is demonstrated through out-of-domain evaluations. The ability to perform well in different domains further strengthens the applicability and effectiveness of CQD^A.

**Weaknesses:**

1. Interpretability of Results: While the paper mentions that the proposed approach provides interpretable answers, it would be beneficial to provide some concrete examples or explanations to illustrate this interpretability.
2. Further Analysis of Training Data Reduction: The paper mentions that CQD^A achieves competitive results with only 1% of the training data. It would be interesting to see a more detailed analysis of the impact of reducing the training data on the model's performance across different query types and datasets.
3. Future Directions: It would be valuable to discuss potential future directions for the research. This could include exploring the applicability of CQD^A in different domains or investigating the scalability of the approach to larger knowledge graphs.

**Questions:**

see weaknesses.

**Limitations:**

Limitations are ignored in this paper.

---

> ### Author Rebuttal · Authors · 2023-08-08
>
> We thank you for your comments and valuable feedback. We would like to address the following points:
> - It would be beneficial to provide some concrete examples or explanations to illustrate this interpretability.
>
> We would like to refer you to our global response, where we clarify why CQD$^\mathcal{A}$ remains as interpretable as CQD. We will provide an additional analysis for this aspect in the camera-ready version.
>
> - Discussion of future work.
>
> In our conclusion, we emphasise that an important direction of future work lies in fundamental enhancements to link prediction methods. With CQD$^\mathcal{A}$, these improvements can be easily adapted for complex query answering in a data-efficient and computationally efficient way. However, we also identify other promising avenues for future exploration. One such area is the significant gap across all methods between EPFO queries and queries containing negations, potentially revealing a core limitation in existing query-answering techniques. Additionally, we propose investigating the specific type of link predictor used in CQD$^\mathcal{A}$, which employs ComplEx+N3 as the default model. This exploration may pave the way for designing link prediction models that generalise better to complex query answering. We will include this discussion in the camera-ready version.

---

> > ### Comment · Reviewer_jfF5 · 2023-08-18
> >
> > Thanks for clarification! But I'm still interested in the impact of reducing the training data on the model's performance across different query types and datasets. The results needn't to be included in the paper. You can comment after the discussion. I'm just curious that whether some types of query data make different contributions to the performance, which may be similar to instruction fine-tuning in NLP.

---

### Official Review · Reviewer_oS1P · 2023-07-07

**Soundness:** 3 good
**Presentation:** 4 excellent
**Contribution:** 2 fair
**Rating:** 5
**Confidence:** 4

**Summary:**

This paper proposes an adaptation model on top of neural link predictors to learn score re-calibration to suit complex query answering task.Shows empirical evaluation on standard benchmark datasets to show its value. One of the benefits is a simple calibration model with lesser training data on complex questions gives better results than ones present in the literature.

**Strengths:**

Paper is well written and easy to follow. Contributions are clearly spelled out.
Experimental results cover complex query answering benchmarks and show the value.

**Weaknesses:**

Contribution seems incremental on top of the existing works. Seems like a small extension to CQD.
Overall results show benefits only on certain query cases compared to state of the art.

**Questions:**

Model is fine tuned with 2i and 3i queries, and test results on those two query patterns don't show any win in two datasets. Any comment on why its happening?

Method seems to get clear benefit on NELL but not on other datasets. any comments on why its happening.

**Limitations:**

Yes

---

> ### Author Rebuttal · Authors · 2023-08-08
>
> We thank you for your comments and valuable feedback. We would like to address the following points:
> - Contribution seems incremental.
>
> Our analysis of CQD points to a fundamental limitation of CQD that is not necessarily trivial to solve if we want to maintain its favourable properties, such as data efficiency and interpretability. We directly tackle the fundamental limitation of uncalibrated scores with a lightweight function that maintains interpretability (due to its linearity), which we argue is not an obvious extension but rather effective compared to significantly more computationally demanding and less data-efficient methods.
>
> - Method seems to get clear benefit on NELL but not on other datasets. any comments on why its happening.
>
> We respectfully disagree with this conclusion. In the case of EPFO queries on FB15k and FB15k-237, CQD$^\mathcal{A}$ remains close to GNN-QE, but CQD$^\mathcal{A}$ outperforms all baselines on all datasets, on queries including negations, all while using 10^3 fewer parameters. In the low-data regime (described in the Data Efficiency subsection), the advantages of CQD$^\mathcal{A}$ over GNN-QE become more pronounced. In additional results (see Table 2 in rebuttal PDF), we have included an additional comparison with BetaE in the low-data regime, where CQD$^\mathcal{A}$ is still producing significantly more accurate results than the baselines.

---

### Official Review · Reviewer_jjs4 · 2023-07-08

**Soundness:** 3 good
**Presentation:** 3 good
**Contribution:** 3 good
**Rating:** 7
**Confidence:** 4

**Summary:**

The paper proposes CQD$^\mathcal{A}$ an adaptive variant of the complex query decomposition (CQD) approach. The authors identify that the neural link predictors used in CQD produce uncalibrated scores that can be in very different ranges of each other. The authors show that CQD$^\mathcal{A}$ can adaptively learn an adaptive function that alleviates this problem.
In an extensive evaluation the authors show that this adaptive calibration imbues CQD with favorable properties such as being more data efficient, robust to OOD and being able to support negation.

**Strengths:**

The paper addresses an important limitation of CQD, wherein the Neural link predictors can produce vastly different ranges of scores leading reduced performance. The authors provide a very simple fix that (inspired from Platt's scaling) allows these scores to be calibrated.
The authors show that this allows the model to boast better results on three benchmarks. The authors also show that this adaptive setting allows the modelling for negations that was previously not feasible.  (**Significance** and **originality**)

The paper is generally well written and easy to follow. (**clarity**)

The authors also provide extensive analysis with regards to their proposed approach which can give useful insights regarding what helps improve performance. (**quality**)




**Weaknesses:**

I think one weakness is that the approach lacks a great deal of novelty. The proposed method is a minor augmentation to an existing approach. However, I still believe that the authors have identified an important problem with CQD and have provided a technically sound solution.



**Questions:**

Perhaps the part about negation could be better explained in the paper.  The authors say that negations are modelled by either a standard or strictly cosine functions. In theory this could be added to vanilla CQD as well. Was this experimented with? Maybe improved results ny CQD$^\mathcal{A}$  over vanilla CQD would bolster the importance of calibration for supporting negations.

**Limitations:**

The authors have not addressed the limitations or potential societal impacts of their work.
 Perhaps the authors could talk about how CQD$^\mathcal{A}$ can produce explainable solutions to complex questions which can create systems that can be verified.

---

> ### Author Rebuttal · Authors · 2023-08-08
>
> We thank you for your comments and valuable feedback. Regarding your questions,
> - Negations are modelled by either a standard or strictly cosine functions. In theory this could be added to vanilla CQD as well.
>
> Even though the link prediction scores are not explicitly calibrated in the original formulation of CQD, we agree that negations can, in principle, be applied with similar functions. Based on your comments, we updated the results in Table 5 to include negations with CQD in our comparison (see Table 1 in the rebuttal PDF), and we find that the model still benefits from an additional calibration via fine-tuning step. This confirms the effectiveness of calibrating the scores for the complex query-answering task.
>
> - How can CQD$^\mathcal{A}$ produce explainable solutions to complex questions which can create systems that can be verified?
>
> We would like to refer you to our global response, where we clarify why CQD$^\mathcal{A}$ remains as interpretable as CQD. We will update the camera ready with additional analysis on this aspect.

---

> > ### Comment · Reviewer_jjs4 · 2023-08-18
> > **After rebuttal**
> >
> > Thanks for the clarifications!

---

### Author Rebuttal · Authors · 2023-08-08

We thank the reviewers for their time and valuable feedback. We appreciate that reviewers acknowledge CQD$^\mathcal{A}$ proposes a novel (jfF5, 5Xze) and technically sound method (jjs4) for data-efficient complex query answering.We have incorporated the provided feedback into our work, including additional experiments (highlighted in red) in the rebuttal PDF.

**Effectiveness of CQD$^\mathcal{A}$ in additional settings**

In the rebuttal PDF, we include additional experiments motivated by the reviewer’s feedback that further indicate the effectiveness of CQD$^{\mathcal{A}}$:
- Table 1 shows the result of vanilla CQD applied directly to queries with negations. We observe that CQD$^{\mathcal{A}}$ still yields more accurate results, indicating that simply normalizing via the sigmoid, or min-max re-scaling, may not be sufficient to solve issues caused by non-calibrated scores, and thus CQD$^{\mathcal{A}}$ significantly benefits from an additional fine-tuning step on the downstream complex query answering task.
- Table 2 includes an additional comparison with BetaE, showing how CQD$^{\mathcal{A}}$ yields significantly more accurate results on the downstream complex query answering tasks.

**Interpretability of CQD$^{\mathcal{A}}$**

Reviewers jfF5 and jjs4 indicated that it is not clear whether CQD$^{\mathcal{A}}$ maintains interpretability. One of the advantages of CQD is the ability to inspect link prediction scores at each step of the query-answering process. This allows to explicitly access and assess the intermediate candidate assignments made by the model for each reasoning step. Our proposed calibration function is monotonic, so the order in the ranking produced by the link prediction scores at each step is preserved and can be interpreted in the same way as in CQD (note that calibration does affect how scores are combined during complex query answering). This would not necessarily be the case if the calibration function were nonlinear. We will clarify this aspect in the camera-ready version.

We provide answers to the remaining questions in individual responses.

---

### Decision · Program_Chairs · 2023-09-21

**Decision:**

Accept (poster)

**Comment:**

The authors propose a method for calibrating neural link predictors that leads to substantial improvements in answering complex questions against a knowledge graph.

All reviewers agree that the paper addresses an important limitation in a good way, which is carefully evaluated and demonstrated to lead to sizable improvements.